



# Benefits and challenges of dynamic sea-ice for weather forecasts

Jonathan J. Day[1], Sarah Keeley[1], Gabriele Arduini[1], Linus Magnusson[1], Kristian Mogensen[1], Mark Rodwell[1], Irina Sandu[1], Steffen Tietsche[2]

[1]European Centre for Medium Range Weather Forecasts, Shinfield Park, Reading, RG2 9AX, United Kingdom
[2][1]European Centre for Medium Range Weather Forecasts, Robert-Schuman-Platz 3, 53175 Bonn, Germany

*Correspondence to*: Jonathan J. Day (jonathan.day@ecmwf.int)

**Abstract.** The drive to develop environmental prediction systems that are seamless across both weather and climate timescales has culminated in the development and use of Earth system models, which include a coupled representation of the atmosphere, land, ocean and sea ice, for medium-range weather forecasts. One region where such a coupled Earth system approach has the
potential to significantly influence the skill of weather forecasts is in the polar and sub-polar seas, where fluxes of heat, moisture and momentum are strongly influenced by the position of the sea ice edge. In this study we demonstrate that using a dynamically coupled ocean and sea ice model in ECMWF Integrated Forecasting System, results in improved sea ice edge position forecasts in the northern hemisphere in the medium-range. Further, this improves forecasts of boundary layer temperature and humidity downstream of the sea ice edge in some regions during periods of rapid change in the sea ice
compared to forecasts in which the sea surface temperature anomalies and sea ice concentration do not evolve throughout the forecasts. Challenges and limitations, such as the quality of ocean and sea ice initial conditions or analyses, and the inability of the coupled system to capture the rate of sea ice concentration change during periods of ice advance and retreat will also be discussed.

## 1 Introduction

Dynamic sea ice and ocean have long been recognised as important components in the Earth System Models used to generate climate projections (Holland and Bitz, 2003; Manabe and Stouffer, 1980) and more recently in seasonal forecasts (Guemas et al., 2016; Koenigk and Mikolajewicz, 2009; Tietsche et al., 2014). This is to meet the societal demand for information on the future state of the sea ice itself (e.g. Melia et al., 2017; Stephenson et al., 2013) and to capture important climate feedbacks
and the remote influence of sea ice on atmospheric circulation (Balmaseda et al., 2010; e.g. Screen, 2017). However, the benefit of sea ice coupling on the timescales relevant for global Numerical Weather Prediction (NWP), i.e. days to weeks, has received less attention.

Until recently it was assumed that sea-ice fields change so slowly that it is acceptable to keep them fixed for the period covered
by global medium-range weather forecasts, so NWP systems typically only included a simple bulk thermodynamic



representation of the sea ice, which enables variations in sea ice surface temperature during the forecast, without the additional complexity of varying the sea ice concentration and thickness (e.g. Mironov et al., 2012). However, this is not the case even for 5-day forecasts in the marginal ice zone, where the total ice cover can change by more than 5% (and 10%) in the transition seasons in the Northern (and Southern) Hemispheres (Keeley and Mogensen, 2018).


The presence of sea ice dramatically influences turbulent exchange at the surface, particularly in winter, when the overlying atmosphere is much colder than the open ocean. As a result errors in sea ice concentration have the potential to degrade the skill of atmospheric forecasts (Jung et al., 2016). For example, during off-ice atmospheric flow situations in winter months (e.g. during Marine Cold Air Outbreaks, MCAOs) the position of the sea ice edge is a strong control on turbulent exchange.

As a result it has been shown in idealised Large Eddy Simulations (LES) experiments that the geometry and position of the sea ice edge strongly influences boundary layer development hundreds of km downstream of the sea ice (Gryschka et al., 2008 and Liu et al., 2006) and as a result can influence the track and intensity of hazardous polar lows (Sergeev et al., 2018) on timescales relevant for short and medium-range NWP.

For these reasons adding sea ice and ocean components to a forecasting system has the potential to increase forecast skill, particularly in locations close to sea ice edge. Therefore, as part of a drive to develop a forecasting system that is seamless across timescales the European Centre for Medium-Range Weather Forecasts (ECMWF) took the pioneering step of coupling sea ice cover and sea surface temperatures (SSTs) between the dynamic–thermodynamic ocean-sea ice model NEMO-LIM2 and ECMWF's Integrated Forecasting System (IFS) for all time ranges, thereby developing the first coupled global medium-

range ensemble forecasting system including dynamic sea ice (Keeley and Mogensen, 2018). Published literature evaluating operational and candidate and coupled NWP systems have been quite broad in scope (Smith et al., 2018; Vellinga et al., 2020) or focused on tropical cyclones (Mogensen et al., 2017). However, little focus has been given to the evaluation of coupled forecast performance in the high latitudes where there is potentially much to gain from coupling the atmosphere with the ocean and sea ice (Jung et al., 2016).


In this study we perform an evaluation of a set of 10-day forecasts with the coupled IFS, comparing them to an equivalent set of uncoupled forecasts, where observed sea ice concentration and SST anomalies from the initial time of each forecast are persisted throughout the forecast range. This enables us to explore three questions relevant to the ongoing development of coupled NWP systems more generally:

1.      On what timescales does dynamic coupling to the ocean-sea ice model produce noticeably improved forecasts of the ice edge?

2.      Does skill in forecasting the position of the sea ice edge improve in all conditions, or during specific episodes?

3.      Is there evidence that the coupling to the dynamic ocean-sea ice has an impact on downstream atmospheric conditions?



## 2 Methods

### 2.1 Experiments

To evaluate the impact of ocean and sea ice coupling on sea ice and atmospheric forecasts in the medium-range, three sets of 10-day forecasts, starting at 0UTC, were run with the ECMWF-IFS for the period DJFM 2017/18. One in which dynamic coupling with sea ice concentration and ocean is switched on (coup-SSTSIC), one atmosphere-only with persisted sea ice concentration and persisted anomaly SSTs (pers-SSTSIC), and one atmosphere-only with updated observed sea ice concentration and SSTs (obs-SSTSIC). These were all run with Cycle 46r1 of the IFS at TCo1279 (~9km) resolution with 137 vertical levels, which is the current resolution of the operational high-resolution 10-day forecasts at ECMWF.

In the uncoupled forecasts (pers-SSTSIC and obs-SSTSIC), sea ice concentration and sea surface temperature (SST) from the Operational Sea Surface Temperature and Sea Ice Analysis (OSTIA, Donlon et al., 2012) is prescribed at the surface. In pers-SSTSIC sea ice concentration and the SST anomalies are persisted from the initial time to the end of the forecast, as was done in ECMWF high-resolution operational forecasts until the IFS Cy45r1 upgrade in June 2018 (the ECMWF Ensemble forecasts have included dynamic sea ice since the Cy43r1 upgrade in November 2016). In this experiment a persisted anomaly approach is used for the SSTs, where the anomaly at the initial time is added to the daily climatology appropriate for each forecast lead time. In the other uncoupled experiment, obs-SSTSIC, the SST and sea ice fields are updated daily throughout the forecast, again using OSTIA. This experiment allows one to assess the maximum potential benefit of correctly representing SSTs and sea ice on atmospheric forecast skill (assuming the evolution of the overlying atmosphere is consistent with the sea ice changes).

The OSTIA fields used in the ECMWF operational analysis (during the period used in this study) are updated daily at 12UTC and fixed until the next day at the same time. The OSTIA analysis fields are based on observations collected during a 36-hour window centred on 12UTC the previous day (Donlon et al., 2012). As a result the sea ice concentration and SST fields used in the pers-SSTSIC and obs-SSTSIC runs are approximatly 36 hours old at the initial time. The sea ice fields in OSTIA are derived from the European Organisation for the Exploitation of Meteorological Satellites (EUMETSAT) Ocean and Sea Ice Satellite Applications Facility (OSI-SAF) sea ice concentration product (Tonboe and Lavelle, 2016), but are interpolated onto the OSTIA grid and adjusted to make the ice concentration consistent with the OSTIA SSTs.

For the coupled forecasts (coup-SSTSIC), the IFS atmosphere is coupled to NEMO (Nucleus for European Modelling of the Ocean) (Madec, 2008) model version 3.4.1 and LIM2 (The Louvain-la-Neuve Sea Ice Model version 2), using the ORCA025 horizontal grid (with a resolution of a quarter of a degree) with 75 levels in the vertical. The IFS makes use of a single executable framework, using a sequential coupling procedure described in Mogensen et al. (2012). Operationally the only sea ice model variable that is coupled to the ocean is the sea ice cover (Keeley et al. 2021 in prep). For the SSTs the atmosphere and ocean are fully coupled in the tropics at all lead-times, but only partially coupled in the extratropics to avoid SST

biases that would degrade forecast skill. In these regions of partial coupling, during the first four days of the forecast, rather than the atmosphere seeing the actual SST field from the ocean model, SSTs from OSTIA are provided at the initial time.

These are then updated by adding the SST tendencies from the ocean model onto the intial field.

In the coupled forecasts, the ocean and sea ice fields are initialised from the ECMWF OCEAN5 analysis (Zuo et al., 2019). OCEAN5 is uses the same NEMO version used in the forecasts. OSTIA sea ice concentration and SSTs are assimilated in it's production in addition to other ocean data sources. The atmosphere of all three experiments is initialised from the ECMWF

operational analysis.

## 2.2 Verification data and metrics

### 2.2.1 Sea ice edge evaluation

We compare the coupled and uncoupled sea ice forecasts using the Integrated Ice Edge Error (IIEE) metric (Goessling et al.,

2016) which measures the skill of forecasts of the ice edge. It is defined as the total area where the forecast and the observations disagree on the ice concentration being above or below a given value, that is, the sum of all areas where the local sea ice extent is overestimated (O) or underestimated (U): IIEE=O+U. Since this metric captures spatial information on the position of the sea ice edge, it is a natural choice for evaluating the impact of including sea ice dynamics over a given region.

An ice concentration of 20% was used as the ice-free/ice-covered threshold in the IIEE calculation, instead of 15% (as used in Goessling et al., 2016), to facilitate a fair comparison between the coup-SSTSIC and pers-SSTSIC forecasts. This is because in the pers-SSTSIC experiment ice concentrations less than 20% were set to zero, as was done in operations at ECMWF prior to the implementation coupled system.

### 2.2.2 verifying analysis

In the IIEE and other metrics, the daily mean sea ice concentration from the forecasts (calculated from 6 hourly fields at 6,12,18 and 24UTC) is compared with the operational OSI-SAF analysis (OSI-401b) for the appropriate day (Tonboe and Lavelle, 2016) to evaluate the sea ice forecast skill. The choice of analysis to use for verification is a complex issue and has received significant attention for atmospheric fields in NWP, where one needs to objectively determine whether one version of a forecasting system is better than another to advance forecast skill (Geer, 2016). There is, however, little guidance for sea ice

evaluation on NWP timescales so some subjective evaluation is performed against Moderate Resolution Imaging Spectroradiometer (MODIS) in Subsection 3.1.





Atmospheric forecast fields are evaluated against the ERA5 reanalysis (Hersbach et al., 2020). Although ERA5 is produced at a lower resolution and with an older version of the IFS than the experiments in this study, ERA5 does not have the 36-hour

lag in the OSTIA SST and sea ice boundary conditions present in the operational analysis and should therefore provide a better estimate of the atmospheric state at the sea ice edge. Therefore the evolution of the sea ice and atmospheric boundary layer should be more consistent than in the operational analysis.

## 3. Results

### 3.1 Sea ice forecast skill

Since the forecast experiments were performed for winter (DJFM), which is a period of ice expansion in the northern hemisphere, persisting sea ice (in the pers-SSTSIC runs), results in negative ice concentration (Fig 1a) and extent (Fig 2) biases which grow with lead-time. The pattern of the bias is more complicated in the coup-SSTSIC (Fig 1d, e & f). The bias is corrected, by construction, in the obs-SSTSIC runs (not shown) and improved relative to pers-SSTSIC in some regions in the coup-SSTSIC runs, since the ice extent is free to increase (or decrease) in the coupled model.


In some regions the coup-SSTSIC forecasts exhibit a positive bias. This is particularly the case in the region north of Svalbard, along the Atlantic coast of North America, and in the Sea of Okhotsk (Fig 1), as can also be seen in the timeseries of northern hemisphere sea ice extent (Fig 2a). The spatial pattern of the bias in the coupled forecasts is consistent across lead times, but the magnitude increases. This suggests that the pattern of the bias, with respect to OSI-SAF, is present in the OCEAN5 analysis,

i.e. already at the initial time of the forecast, and is inherited from biases in the background forecasts from the ice-ocean model, which advances the ice edge too rapidly, consistent with the findings of Tietsche et al. (2015).

Subjective evaluation of the sea ice edge position in the OSI-SAF and ECMWF-OCEAN5 analyses against True Colour and Corrected Reflectance images from the Moderate Resolution Imaging Spectroradiometer (MODIS) for selected forecast dates

confirms the day-0 differences (see Fig 3). In the MODIS Corrective Reflectance product (shown in Fig 3b, d & f) snow and ice appear bright red. Thick ice and snow appear vivid red (or red-orange), while ice crystals in high-level clouds will appear reddish-orange or peach. Open ocean appears dark but small liquid water drops in clouds appear white. This strong contrast between the red (sea ice) and the dark (open ocean) shades in the images can be used to infer the position of the sea ice edge. Overall, the OSI-SAF sea ice edge is more consistent with the MODIS images than the OCEAN5, which further justifies the

use of OSI-SAF as the reference data for the sea ice verification in this study. The largest difference between OSI-SAF and OCEAN5 in the Nordic Seas are in the region north and west of Svalbard (Fig 1d). A large area of open water is present in this region and OCEAN5 overestimates the extent of the sea ice (see the region annotated "NS" in Fig 3a), with the OSI-SAF ice edge being a closer match to the MODIS image (i.e. comparing the line contours to the edge of the red ice-covered area in Fig 3b). In the Western Atlantic the differences between OSI-SAF and OCEAN5 are largest in the regions adjacent to the east



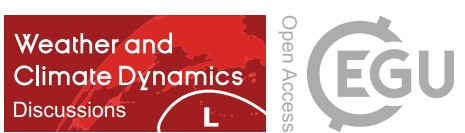

and west coast of Greenland, where OSI-SAF is more extensive, stretching further south along both coastlines than in OCEAN5 (Fig 3c & d). On the 3-3-2018 there is a clear discrepancy along the eastern coast of Greenland for example (in the region annotated "EG" in Fig 3c), where ice is completely missing in the OCEAN5, but present in MODIS and OSI-SAF. There is a similar situation in the northern part of the Sea of Okhotsk (in the region indicated with the initials SO in Fig 3e), where OCEAN5 is more extensive than the OSI-SAF fields. The truth is harder to see for the bias in West Greenland, indicated by

WG in Fig 3c, due to the high level of cloud cover in the image shown and in others that were inspected on different dates that are not shown.

In Fig 3 the OSI-SAF fields from the previous day are plotted in orange, to provide an indication of how large the difference one would expect from the OCEAN5 field simply due to a lag in the SIC field used in the assimilation. The similarity between

the OSI-SAF fields from one day to the next and the difference between these and the OCEAN5 fields, suggests that errors in the OCEAN5 are systematically biased in these regions, rather than due to a simple lag in the availability of sea ice concentration data for use in the production of the ocean analysis mentioned in subsection 2.2.1. The fact that biases in these regions show pronounced growth with lead time (see Fig 1 and Fig 2) further supports the idea that these biases are inherited from the ice-ocean model via the background fields used in the assimilation process. This can be seen most clearly in the

timeseries of sea ice extent for the sea of Okhotsk where there is a systematic positive bias in the OCEAN5 analysis (red dots) compared to OSI-SAF (black dots), which grows with leadtime during the forecasts (orange lines, see Fig 2d).

For objective scoring of the forecasts, we calculate the IIEE with respect to OSI-SAF, which provides an integrated measure of forecast performance for the position of the sea ice edge in a given region. It provides a detailed picture of how accurate the

forecasts are in predicting which grid boxes are covered with sea ice by taking misplacement of the ice as well as the total area of ice into account (Goessling et al., 2016). It was calculated for the northern hemisphere as a whole and also for the Nordic Seas (20W-60E, 65-83N), Western Atlantic/Labrador Sea region (62-30W, 55-70N) and the Sea of Okhotsk (135-157E,45-62N). These regions are shown in the coloured boxes in Fig 1c & f.

IIEE increases with lead-time for both pers-SSTSIC and coup-SSTSIC, with pers-SSTSIC generally increasing more rapidly. The cross-over between pers-SSTSIC and coup-SSTSIC for any given region largely depends on the differences in initialisation in that region. In the Northern Hemisphere and Nordic Seas region coup-SSTSIC shows an improvement early in the forecast. However, the coup-SSTSIC forecasts have a larger IIEE in the Labrador and North Atlantic region at all lead times and in the Sea of Okhotsk until day 6.


It is interesting to note the large initial (day-0) IIEE in both pers-SSTSIC and coup-SSTSIC experiments (see also Zampieri et al., 2018). Errors in the pers-SSTSIC forecast at day-0 are mainly due to the lag in the time at which sea ice fields are available to the operational analysis (see section 2.1 for explanation) and partly due to small differences between the OSI-SAF fields,





used for evaluation, and OSTIA fields, used as boundary conditions to the model, due to interpolation and adjustment for
consistency with the SSTs used in the OSTIA product.

Initial errors in coup-SSTSIC are inherited from the OCEAN5 analysis as indicated by the similarity between the coup-SSTSIC
IIEE at day-0 and the IIEE of OCEAN5 w.r.t to OSI-SAF (i.e. comparing the orange and red curves in Fig 4). The causes of
errors in OCEAN5 were already discussed above, however it is worth noting that the coup-SSTSIC day-0 error for the Northern
Hemisphere, w.r.t OSI-SAF, is more than 50% of the IIEE of the coupled forecast at day 9. It is also striking that the error in
the analysis, expressed as the IIEE of the OCEAN5 fields, is larger than the difference in IIEE between the coup-SSTSIC and
pers-SSTSIC forecasts (Fig 4) which suggests that using OCEAN5 instead of OSI-SAF for verification would dramatically,
and erroneously, change the outcome of the evaluation.

### 3.2 Improved sea ice forecasts during periods of ice advance and retreat

In this section we investigate whether the difference in skill between pers-SSTSIC and coup-SSTSIC is consistent across the
period of study or is larger during specific episodes. Timeseries of the IIEE  at day-3 (Fig 5) and day-9 (Fig S1) show that in
general the pers-SSTSIC forecasts are more variable in all regions at both leadtimes, except for the Sea of Okhotsk at day-3.
This high variability of the error  shows that persistence is at times a very good forecast, but that at times it is very poor.
Decomposing the IIEE of the pers-SSTSIC forecast into the absolute extent error (AEE) and misplacement error (ME)
(following Goessling et al 2016) allows us to investigate this further.

On average the AEE and ME contribute roughly 50% each (Fig S2) during the period investigated (DJFM), although this ratio
may depend on the season. However, further inspection of the timeseries reveals that situations where persistence is a poor
forecast, tend to be where the AEE, rather than ME, is large for the persistence forecast (Fig S2), with the AEE explaining
~2/3 of the variance in the IIEE in each region. This shows that the reason that persistence forecasts are particularly poor in
these episodes is because rapid expansion and contraction of the area covered with ice is missed.

These periods of rapid sea ice change are exactly the situations where one would expect the dynamic coupling to  add the most
value. This can be seen most intuitively from Figure 2, which shows that pers-SSTSIC is a particularly poor forecast during
periods of rapid extent change. During periods where the observed extent increases (decreases) the persistence forecast is
biased low (high). As the sea ice in the coup-SSTSIC forecasts is dynamic, the model is able to follow these variations seen in
the observations. As a result, the reduction in IIEE ($IIEE_{coup-SSTSIC}-IIEE_{pers-SSTSIC}$) is largest during periods of rapid ice extent
change. There is a significant correlation between $IIEE_{coup-SSTSIC}-IIEE_{pers-SSTSIC}$ and the magnitude of the observed change in
ice extent between the initial and verification times in the Nordic, Labrador, and Okhotsk seas at day-3 (r=-0.48, -0.47, and –
0.35 respectively, all highly significant) and day-9 (r=-0.54, -0.62, and –0.61 respectively, all highly significant) (see Fig 6).

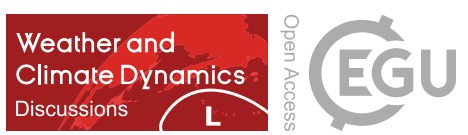

Indeed, even for lead-times where IIEE$_{coup-SSTSIC}$ is larger than IIEE$_{pers-SSTSIC}$ on average (e.g. day-3 in the Sea of Okhotsk: Fig 4d) events with exceptionally large changes in extent are better forecasted by coup-SSTSIC (see e.g. Fig 6e).

Such rapid changes either involve an advancing or retreating sea ice edge. Splitting the forecast dates into terciles, based on
the observed change in sea ice extent over the first 3 days of the forecast, and compositing the initial conditions demonstrates that mean north easterly flow in that region is associated with advances in sea ice cover. During such situations southward advection of the sea ice due to anomalous wind stress and antecedent conditions for thermodynamic growth lead to a positive 3-day change in sea ice concentration (Fig 7). Comparison of this composite with the forecast 3-day change in concentration shows that the forecast does not increase as rapidly as seen in observations. A similar picture can be seen in the other sectors
(not shown). Note that the larger IIEE improvement in ice advance cases compared to ice retreat cases likely reflects a positive bias in the rate of ice growth, which tends to favour performance during ice advance.

### 3.3 Potential benefits of ice-ocean coupling on atmospheric forecasts

As air is advected across a boundary between distinct surface-types, such as between sea ice and open water (as in the situation
shown in Fig 6), its properties are modified as it adjusts to the new set of boundary conditions (Oke, 1987). This modification begins at the surface and is propagated upwards through turbulent diffusion. The layer of air whose properties have been affected by the new surface is called an *internal boundary layer* and its depth grows with distance downstream of the boundary between the media, known as the *leading edge*, which in our case is the edge of the sea ice. Here we will focus on variations in the sea ice edge due to their large spatial scale (and therefore importance to global NWP), however air-mass advection
between ice covered and open water regions in the form of leads and polynyas within the ice are also known to strongly affect air-mass properties (Moore et al., 2002; Pinto and Curry, 1995).

Two such idealised situations are represented in schematics 8a and 8b. Fig 8a shows a situation with off-ice-flow, typically seen during Marine Cold Air Outbreaks, where a cold polar air mass is advected across the sea ice edge and over the open
ocean. Since this air is much colder and drier than conditions at the surface of the open ocean upward sensible and latent heat fluxes are induced which act to warm and moisten the internal boundary layer. Conversely, Fig 8b shows an idealised on-ice-flow situation where a marine air mass is advected over the sea ice. Since this air mass is much warmer and more humid than conditions at the surface, anomalous downward sensible and latent heat fluxes are induced which act to cool and dry the internal boundary layer.


This link between the position of the sea ice edge and the development of the downstream boundary layer is a potential source of error in forecasts where the sea ice is persisted from the initial analysis, as in pers-SSTSIC. Because the sea ice concentration is itself modified by anomalous surface wind stress and surface energy fluxes, during the situations described above, the





position of the *leading-edge* changes over time, influencing downstream boundary layer development. In particular, during
persistent off-ice flows, the sea ice edge advances to cover more of the open ocean (see Fig 8c). As a result, the polar air meets
the open ocean further to the south, and the maximum in the turbulent fluxes is shifted further to the south. During persistent
on-ice flows the opposite occurs, with the sea ice edge retreating to expose more open water (Fig 8d). As a result the sea ice
dynamics modify the point at which the air-mass transformation begins to occur. The impact that the ice dynamics can have
on the internal boundary layer development and on the turbulent fluxes can be seen by comparing with Fig 8a and 8b, which
show an idealised situation where the sea ice has been fixed, with Fig 8c and 8d respectively, which show the same situation
but where the sea ice has been allowed to evolve dynamically. The difference in the position of the internal boundary layer is
shown in faded hues.

The features described above can be clearly seen in the composites of mean temperature and specific humidity forecast errors
during periods of ice advance (Fig 8, 9 and 10) and retreat (Fig S3, S4 and S5) in each of the 3 regions (shown in Fig 1c) from
the pers-SSTSIC forecast set during DJFM. The dates used in these composites are those for which the 3-day change in sea
ice extent is in the top or bottom tercile, as used in Fig 6.

The negative bias in ice concentration in pers-SSTSIC, during periods of ice advance, goes hand in hand with positive
temperature and specific humidity bias around and to the south of the sea ice edge (subfigures a, d & g in Fig 9, 10 and 11).
The coup-SSTSIC and obs-SSTSIC forecasts have higher sea ice concentrations than the pers-SSTSIC forecasts (Fig 9, 10 &
11 b & c) and, as a result, the turbulent heat flux (THF) is reduced in those regions (Fig 9, 10 & 11 e,f,h & g). Because the
area south of the sea ice edge is a local maximum of THF field and the ice edge is further south in these simulations compared
to the uncoupled runs, the THF is increased somewhat to the south resulting in a dipole in the heat flux field (i.e. red dashed
contours to the south of the blue). This effect was also observed in studies with more dramatic changes in sea ice (Day et al.,
2012; Deser et al., 2010).

During periods of ice advance the obs-SSTSIC and coup-SSTSIC are cooler, and generally dryer than pers-SSTSIC in the
region downstream of the sea ice edge (Fig 8, 9 and 10 e, f, h and i), this difference is larger where the difference in sea ice
concentration is greatest, but extends some ~800km downstream to the south of the region where the sea ice has changed. To
put into context, the maximum 925hPa specific humidity difference relative to pers-SSTSIC is perhaps 0.2g/kg northeast of
Svalbard. At day-3 in that region the climatological humidity is about 2.5g/kg and the RMSE is about 0.3g/kg (not shown). So
dynamic ice is important in terms of 925hPa specific humidity error (i.e. around 10% of the total humidity).

This is consistent with previous modelling studies which have indicated sensitivity to the nature and position of the sea ice
edge/marginal ice zone as similar distance downstream using LES experiments (e.g. Gryschka et al., 2008; Liu et al., 2006),
although clearly cooling of SST will also play a role. Note that near and downstream of the sea ice the coup-SSTSIC forecast




is cooler and dryer than the obs-SSTSIC run due to having higher sea ice concentrations. These are at least partly associated with a positive bias in sea ice concentration in the Nordic sector (Fig 1). Further, biases in sea ice in the coupled model SE of

Greenland and in the Sea of Okhotsk already discussed, go hand in hand with temperature and humidity biases (Figs 8, 9, S3 and S4). This shows the potential for errors in the sea ice to influence the atmospheric fields.

Conversely, during periods of ice retreat in the Nordic Seas (which corresponds to on-ice flow), a positive bias in the uncoupled runs goes hand in hand with negative temperature and specific humidity biases over the sea ice (Fig S3 a,d &g). The coup-

SSTSIC and obs-SSTSIC forecasts a have lower sea ice concentrations than the uncoupled and as a result the THF is higher in those regions (Fig S3 b & c). Compared to the uncoupled forecast, the prescribed and coupled forecasts are warmer, and more humid over the sea ice downstream of the changes in the sea ice edge (Fig S3 e, f, h and i). However, the response of the atmosphere and turbulent exchange to including the evolution of the sea ice is much more modest in these situations. This is consistent with the findings of Blackport et al. (2019) who argue that due to the orientation of the turbulent fluxes, the influence

of the sea ice position on the atmosphere is much more modest during such situations. A similar picture can be seen in the Labrador Sea and Sea of Okhotsk (Fig S4 and 5).

## 4.Conclusions

A set of 10-day coupled atmosphere-ocean-sea ice forecasts with the ECMWF forecasting system have been evaluated and

compared with uncoupled forecasts with both persisted and updated ocean and sea ice surface fields to determine the benefits of dynamic sea ice coupling for medium-range NWP.

Overall, coupled atmosphere-ocean-ice forecasts with the IFS improve forecasts of northern hemisphere sea ice edge compared to persistence, although some regions see a degradation. Differences in the sea ice concentration fields in the ECMWF ocean

and sea ice analysis, compared to OSI-SAF, suggest that errors in the initial-analysis are a large contribution to sea ice edge errors in the medium-range forecasts. Subjective evaluation of the analysis fields against MODIS data suggests that the OSI-SAF is more consistent with the ice edge in the regions considered in this paper than the ECMWF ocean and sea ice analysis, so evaluation against OSI-SAF provides a more robust evaluation of forecast performance than comparing against ECMWFs own analysis at the present time. This is partly due to bias inherited from the ocean-sea ice model and partly because the

ECMWF operational analysis sees OSI-SAF/OSTIA sea ice conditions that are roughly 36-hours old. The lag could be addressed by assimilating swaths as they become available, rather than using a 24hr composite product.

Further, it is well known that there is significant variation in the sea ice concentration products produced from passive microwave instruments due to the use of different algorithms and these variations mainly occur near the edge of the sea ice

(Meier, 2005). This is one of the reasons to use the IIEE metric (Goessling et al., 2016) for evaluation. Ice concentration retrieval algorithm comparisons have tended to focus on uncertainties in long-term trends (Andersen et al., 2007) and



evaluation of large-scale statistics for climate models (Notz, 2014), rather than on the day-to-day variations at the spatial scales important for NWP applications discussed here. As a result there is limited guidance for how to perform sea ice evaluation for NWP. One route forwards may be to use ice fraction derived from the Advanced Microwave Scanning Radiometer 2 (AMSR2) (Spreen et al., 2008), which provides a higher resolution picture of the sea ice edge more consistent with mesoscale meteorology (Renfrew et al., 2021). However, more guidance on the pros and cons with a focus on variations on day-to-day changes from the remote sensing community would be very useful.

A decomposition of the IIEE metric into the absolute extent error and misplacement error terms showed that persisting sea ice concentration will lead to particularly large ice edge error during periods of rapid ice advance and ice retreat. It was also shown that these are exactly the episodes during which the coupled forecasts add the most value, i.e. the IIEE of the coupled forecasts (coup-SSTSIC) is most reduced compared to the persistence forecast (pers-SSTSIC). Compositing analysis of atmospheric fields demonstrate that periods of ice retreat and ice advance correspond to anomalous "on-ice" and "off-ice" wind patterns respectively.

Interestingly, it is during such "on-ice" and "off-ice" flow situations, where the position of the sea ice is expected to exert a controlling influence on atmospheric boundary-layer development. Investigation of atmospheric forecast errors during these periods shows that errors in the position of the sea ice edge can lead to errors in lower tropospheric temperatures hundreds of km downstream of the ice edge and that such errors were present in forecasts with persisted sea ice (pers-SSTSIC) due to missing sea ice dynamics. Using a set of experiments where observed sea ice conditions are updated as the forecast evolves (obs-SSTSIC) we demonstrate that correctly capturing the evolution of the sea ice in such conditions can reduce forecast errors in these situations. Further, in regions where the coupling improves forecasts of ice edge position, boundary layer properties are also improved. The opposite is true in regions where forecasts of ice edge position are degraded, such in the Sea of Okhotsk.

These results highlight the potential benefits in weather forecasts that could be gained from improving the coupled atmosphere-ice-ocean system, but also the risks of degrading weather forecast performance by introducing errors via the coupling. Such trade-offs will need to be considered when considering future upgrades, but also when deciding whether to go to a coupled system for the next iterations of the ECMWF reanalysis series, where coupling is now a viable option (Laloyaux et al., 2016). That said, coupled assimilation approaches have shown promise to improve the quality of ocean-sea ice analyses, such as the use of weekly coupled data assimilation, where ocean and atmosphere data assimilation are run together, rather than separately, resulting in more consistency between the atmosphere and ocean initial conditions. This looks to be particularly promising in the Baltic, where the proximity of sea ice to land and low ice fractions causes issues for passive microwave sea ice concentration retrievals (Browne et al., 2019).

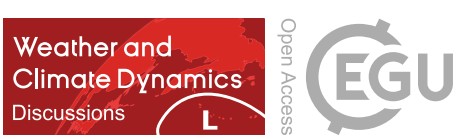

In terms of limitations, this study only considered forecasts of the sea ice edge and its importance for the heat and moisture exchange. Other properties, such as surface roughness variations with ice concentration (Elvidge et al., 2016) can have an important influence on the circulation (Renfrew et al., 2019). Similarly, we did not consider the importance of linear kinematic features or leads in the sea ice which are also potentially important for boundary-layer development. Such features emerge from sea ice models with isotropic viscous-plastic rheologies, like LIM2, at high resolution (i.e. 4km, Hunke et al., 2020).

Such resolutions are potentially within the envelope for future operational medium-range forecasts and are potentially predictable in the medium-range (Mohammadi-Aragh et al., 2018), however, how to evaluate these features and what level of importance they have for medium-range weather forecasts are open questions.

**Data availability**

The experiments described in Table 1 are openly available from the ECMWF Meteorological Archival and Retrieval System (MARS) and published under the following dois:

- pers-SSTSIC: doi: 10.21957/4vw1-0f68
- coup-SSTSIC: doi: 10.21957/xbe4-6v10
- obs-SSTSIC: doi: 10.21957/4r57-jb72


**Author contribution**

J. D., S. K., G. A., L. M., I. S., K. M., conceived and planned the experiments. G. A. and J. D. carried out the experiments. J. D. took the lead on the analysis and writing of the manuscript with frequent input from S. K.. M. R. and S. T. helped J. D. with the evaluation of experiments. All authors contributed to the interpretation of the results, provided critical feedback and helped
shape the research, analysis and manuscript.

**Competing interests**

The authors declare that they have no conflict of interest.

**Acknowledgements**

The work described in this article has received funding from the European Union's Horizon 2020 Research and Innovation programme through grant agreement No. 727862 APPLICATE. The content of the article is the sole responsibility of the author(s) and it does not represent the opinion of the European Commission, and the Commission is not responsible for any use that might be made of information contained. This is a contribution to the Year of Polar Prediction (YOPP), a flagship activity of the Polar Prediction Project (PPP), initiated by the World Weather Research Programme (WWRP) of the World
Meteorological Organisation (WMO). We acknowledge the WMO WWRP for its role in coordinating this international research activity. The authors thank David Richardson for his comments on the manuscript.





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

**Table 1 configuration of experiments**

| Experiment name | Experiment type | SST/sea ice updated | Source of SSTs/ sea ice |
|---|---|---|---|
| pers-SSTSIC | Uncoupled | No | OSTIA |
| obs-SSTSIC | Uncoupled | Yes | OSTIA |
| coup-SSTSIC | Coupled | Yes | OCEAN5 |







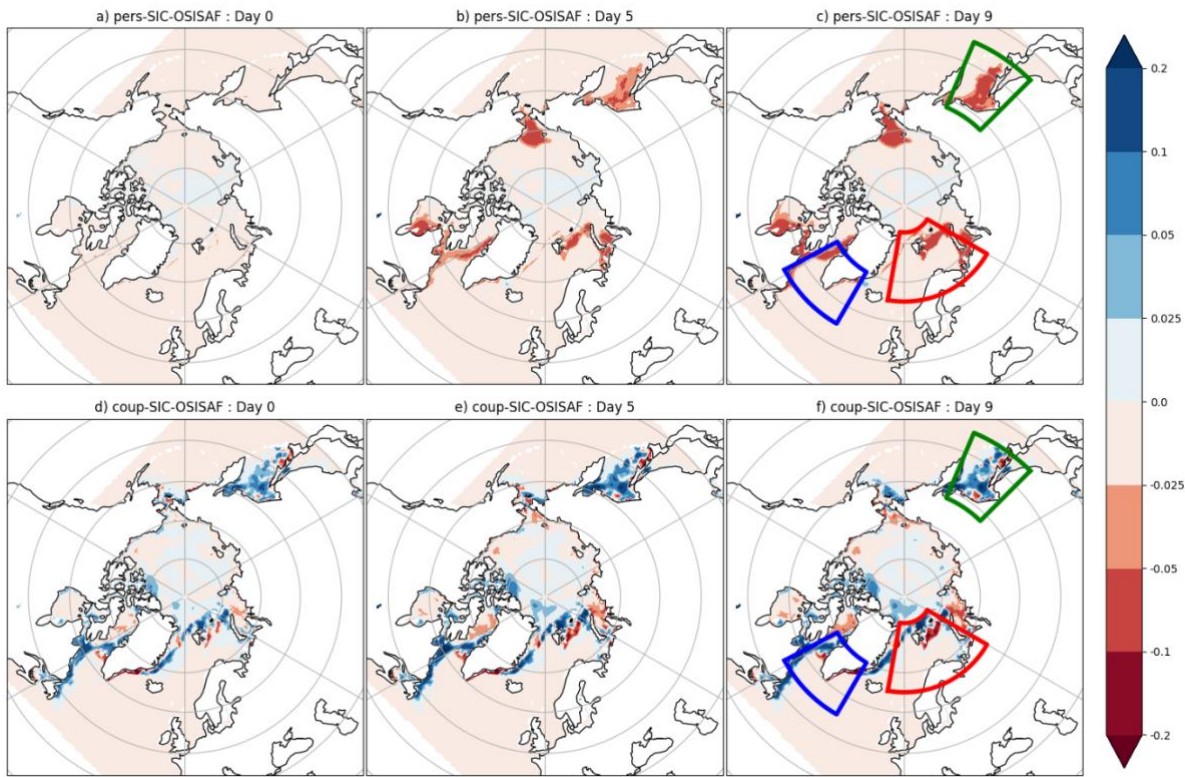

**Figure 1: daily mean sea ice concentration bias, relative to OSI-SAF for the persisted forecasts, pers-SSTSIC, (top) and the coupled**
**forecasts, coup-SSTSIC, (bottom) at a lead time of 0 (a,d), 5 (c,e) and 9 (e,f) days. The Nordic Seas (red), Labrador and North**
**Atlantic region (blue) and Okhotsk Sea (green), are highlighted by the coloured boxes in the right hand column.**

**Figure 2: timeseries of sea ice extent in the northern hemisphere (a), Nordic Seas region (b), Labrador and North Atlantic (c) and the Sea of Okhotsk (d). The regions are shown in Fig 1. Daily mean sea ice extent for each 10-day forecast and each analysis product is plotted with day-0 indicated by a dot marker.**



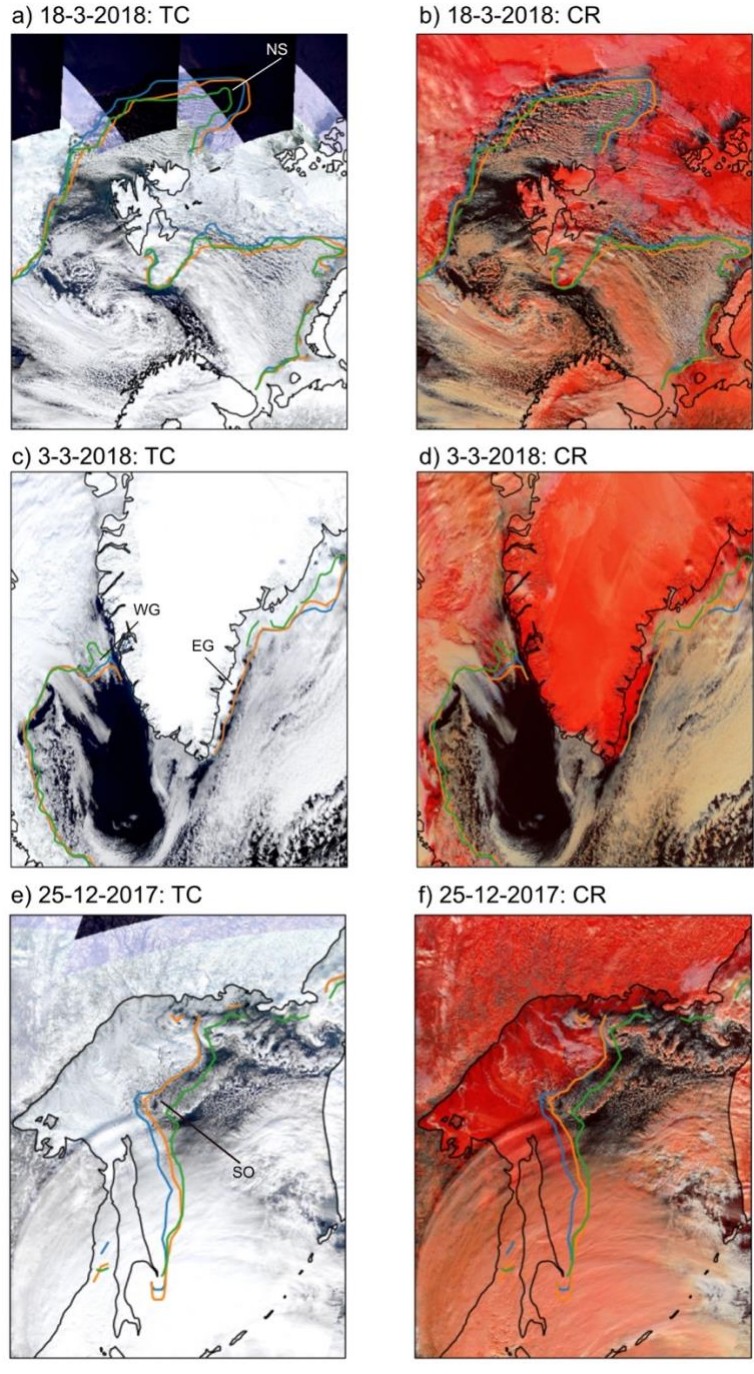

**Figure 3. MODIS True Colour (left) and corrected reflectance (right) images for the Nordic Seas on 18-3-2018 (top) and Labrador and North Atlantic region on 3-3-2018 (middle) and Sea of Okhotsk on 22-12-2017 (bottom). The 0.2 ice fraction contour from the OSI-SAF (blue), OSI-SAF-1 day (orange) and ECMWF-OCEAN5 (green) are overlayed. These regions are shown by the coloured boxes in Fig 1c and f. Areas of large discrepancy between the OCEAN5 analysis and MODIS images are annotated.**


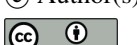


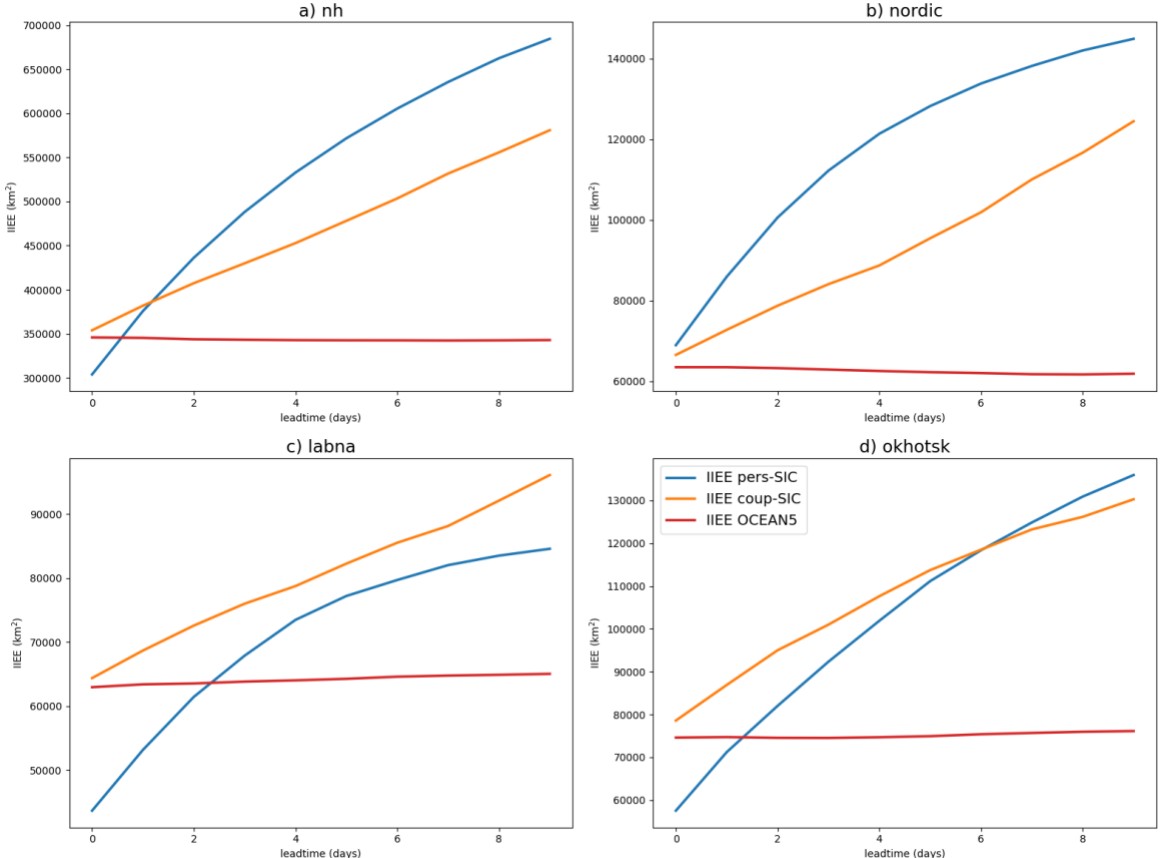

**Figure 4: Average Integrated Ice Edge Error (IIEE) over all forecasts plotted against lead-time, for the whole northern hemisphere (a), the Nordic Seas (b) and the Labrador/East Atlantic (c) and the Sea of Okhotsk (d) regions. These regions are shown in Fig 1c & f.**






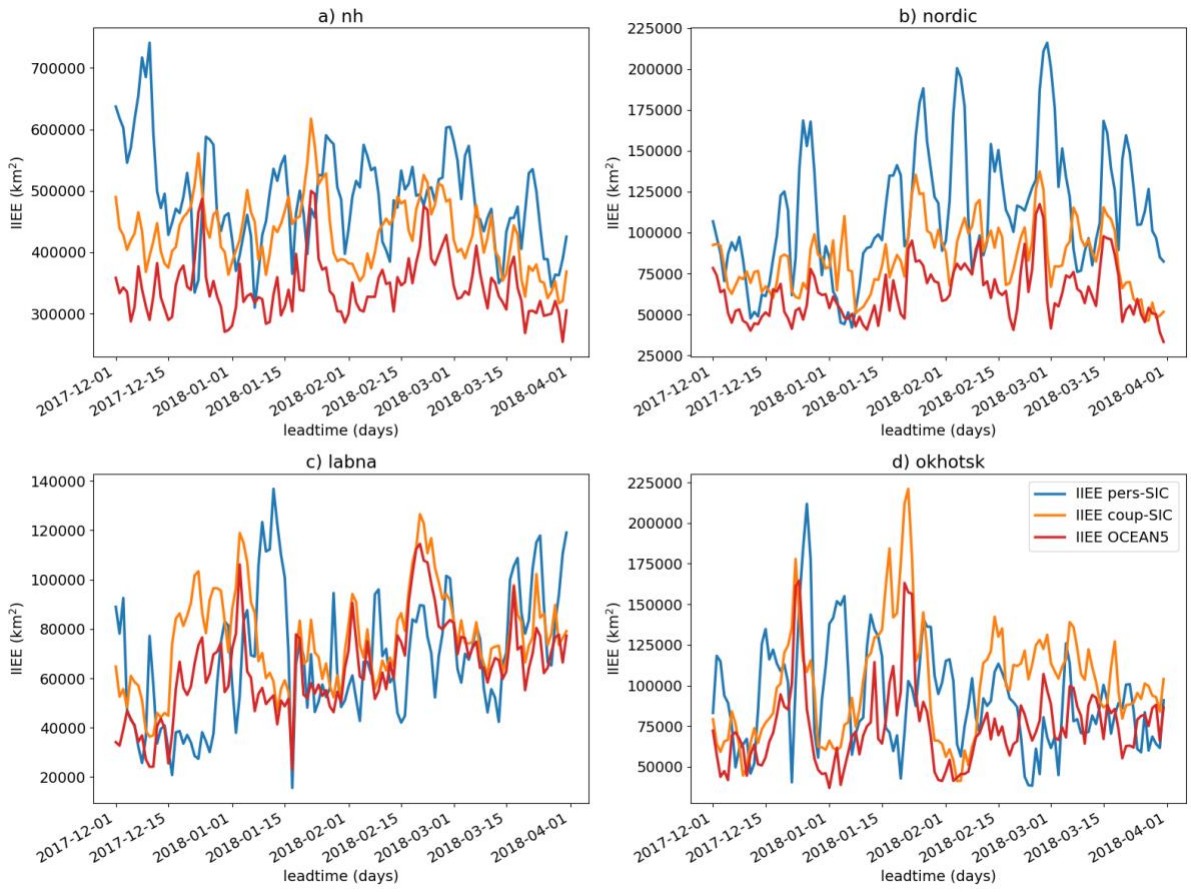

**Figure 5 timeseries of day-3 Integrated Ice Edge Error (IIEE) of all forecasts for the whole northern hemisphere (a), the Nordic Seas (b) and the Labrador/East Atlantic (c) and the Sea of Okhotsk (d) regions. These regions are shown in Fig 1c & f.**




**Figure 6: Scatter plot of difference in IIEE between coup-SSTSIC and pers-SSTSIC forecasts at day 3 (left) and day 9 (right) and the change in observed ice extent between the initial time and the verification time for the Nordic Seas (top row) and the Labrador/East Atlantic (middle row) and the Sea of Okhotsk (bottom row) regions. These regions are shown in Fig 1c & f. Colours correspond to the lowest (green), middle (blue) and highest (red) terciles of ice extent change.**




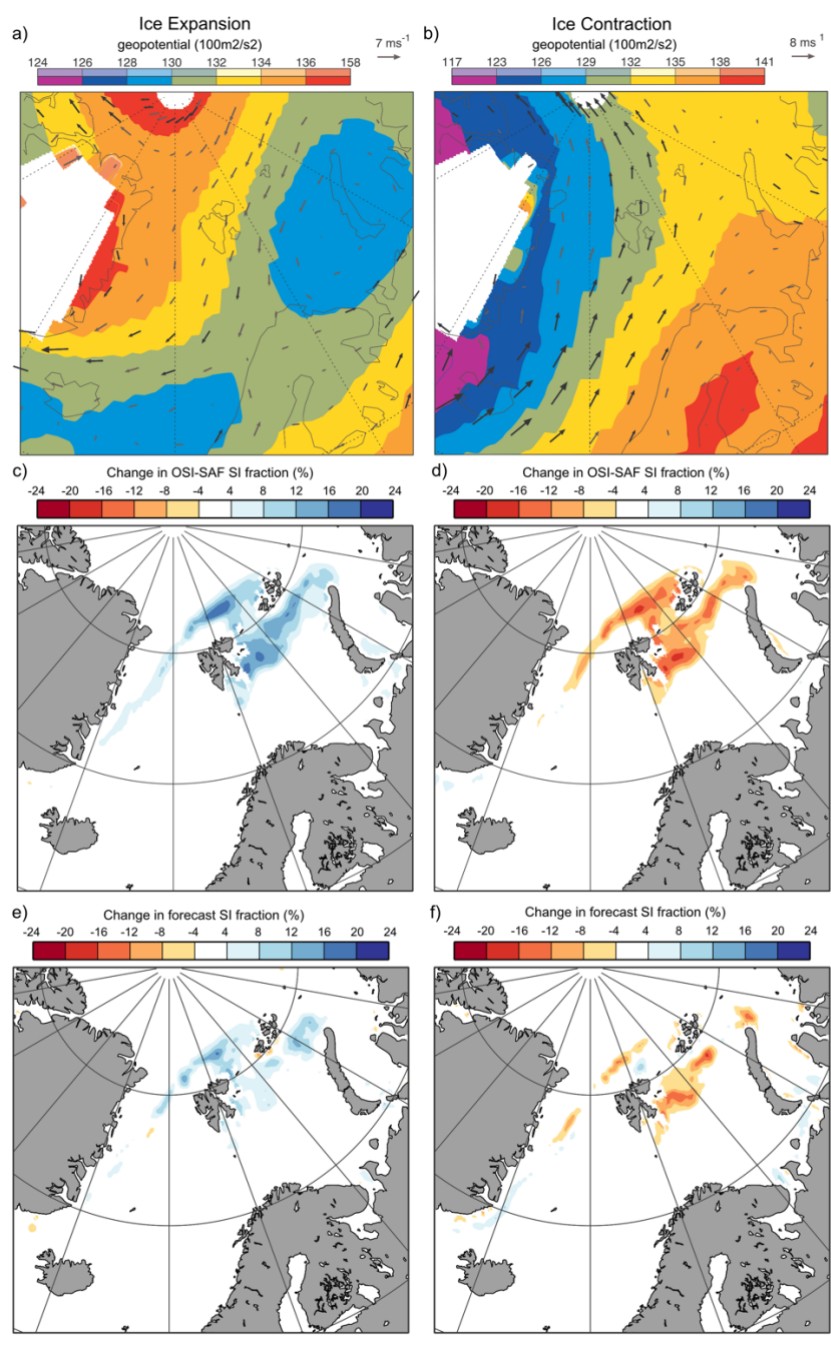



**Figure 7. Composite 850hPa geopotential and vector winds for ice advance (a) and retreat (b) cases selected based on the upper and lower tercile the of 3-day SIE change in the Nordic seas (highlighted in Fig S2). C and d, as a and b but for the 3-day change in observed SIC, e and f, as c and d but for the coupled forecast.**

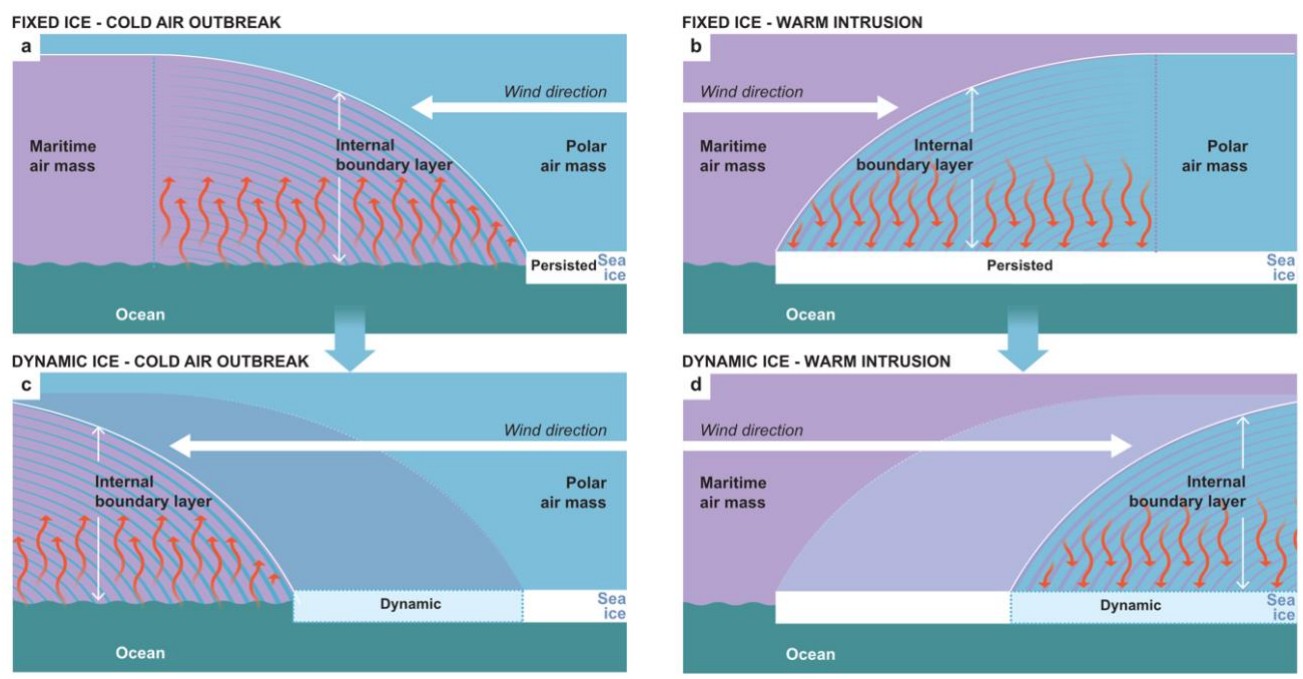


**Figure 8: schematic showing the of impact of dynamic sea ice on turbulent exchange and atmospheric boundary layer development during cold air outbreaks (a and c) and moist intrusions (b and d).**



**Figure 9: Composite of forecast error during periods of ice advance in the Nordic seas: Day-3 sea ice concentration bias along with composite-mean winds at 925hPa (a), T+72 925hPa temperature and horizontal wind bias (d) and specific humidity and horizontal wind bias (g) for the pres-SIC forecasts and the change in the bias for the obs-SSTSIC (b, e and h) and coup-SSTSIC (c, f and i) with respect to the pre-SIC forecasts. The change in the turbulent heat flux (sensible+latent accumulated between T+48 and T+72) is shown in the blue and red contours in b and c (anomalies are positive downwards). In panels (d)-(i), saturated colours indicate mean differences that are statistically significant at the 5% level.**



**Figure 10: Composite of forecast error during periods of ice advance in the Labrador Sea and Baffin Bay: Day-3 sea ice concentration bias along with composite-mean winds at 925hPa (a), T+72 925hPa temperature and horizontal wind bias (d) and specific humidity and horizontal wind bias (g) for the pres-SIC forecasts and the change in the bias for the obs-SSTSIC (b, e and h) and coup-SSTSIC (c, f and i). The change in the turbulent heat flux (sensible+latent accumulated between T+48 and T+72) is shown in the blue and red contours in b and c (anomalies are positive downwards). In panels (d)-(i), saturated colours indicate mean differences that are statistically significant at the 5% level.**





**Figure 11: Composite of forecast error during periods of ice advance in the Sea of Okhotsk: Day-3 sea ice concentration bias along**
**with composite-mean winds at 925hPa (a), T+72 925hPa temperature and horizontal wind bias (d) and specific humidity and**
**horizontal wind bias (g) for the pres-SIC forecasts and the change in the bias for the obs-SSTSIC (b, e and h) and coup-SSTSIC (c,**
**f and i). The change in the turbulent heat flux (sensible+latent accumulated between T+48 and T+72) is shown in the blue and red**
**contours in b and c (anomalies are positive downwards). In panels (d)-(i), saturated colours indicate mean differences that are**
**statistically significant at the 5% level.**
