# Peer review of "Benefits and challenges of dynamic sea-ice for weather forecasts"

_Weather and Climate Dynamics, 2022_

## Author Comment (AC1)

Response to reviewers

We would like to thank the reviewers for their positive and thoughtful comments on the manuscript, which have helped us improve it. Although none of the recommendations resulted in major changes, we have carefully been through and edited the text and figures. The main edits to the text were to make the links between the results subsections clearer and bringing some of the important results and recommendations from the results sections into the abstract and conclusions & discussion section following their suggestions. The discussion of the causes of ocean analysis errors was given its own section (3.2) to give it more prominence in the manuscript.

In the following, comments from the reviewers are stated in black font and the response is written in blue. Passages of modified text from the manuscript are indicated by in blue italics and quotation marks. Where line numbers are used to refer to changes, they correspond to the position of a change in the tracked changes version of the revised manuscript (note that line numbers in the tracked changes and clean pdf versions differ).

**General Comments**

(1) The overall benefits of a dynamic sea ice are clearly evident and are nicely illustrated in Figures 1, 2 (which is a striking illustration) and 4. However, these general (and seasonally averaged) plots do show some caveats. Fig 4 illustrates that over the first day, the persistent forecast has lower IIEE than the coupled forecast for the northern hemisphere, and that this is always true in the Labrador Sea region, while in the Sea of Okhotsk it is true until around day 6. I suspect the reason persistence is better for these seas is that they are relatively small and enclosed regions, with sea-ice that advances/retreats when the winds are along the sea, thus pick out the advancing/retreating problems discussed later. These findings are noted briefly in section 3.1 (e.g., L90-95), but I think further discussion is really needed in section 3.1. This is explored a bit in section 3.2, where I think Fig 6 is used to explain that IIEE and changes in ice concentration are related (especially so for the more enclosed seas), but this is not very well linked back to the key figures of 2 and 4. I suggest the authors work on improving the links between Figs 2-4 and Fig 6-7 and explaining the different qualitative results of Fig 4.

We agree with the reviewer that geographic variations in persistence play a role, but also regional differences in biases and initial errors in the coupled model.

We have added the following at line 191 and made other small edits in the paragraph to reflect this: "*If, and at what lead-time, the coupled forecast becomes more skilful depends on both the size of the initial error in coup-SSTSIC and the rates of error growth of both the coupled model and a persistence forecast. All these factors are regionally dependent: for example, in the Sea of Okhotsk there is a large initial bias in ice extent which grows with lead-time, leading to worse performance than persistence during the first six days. In contrast the coup-SSTSIC initial error in the Nordic Seas is smaller than pers-*

*SSTSIC and biases are more modest and so performance is better than persistence from day-0. The skill of a persistence forecast will also vary from region to region based on the local ice dynamics."*

We have also added some text to, what is now, Section 3.3 to link figs 6 and 7 back to figure 2.

(2) An interesting fact is noted with regard to Fig 4, that the 'initialisation error (IIEE)' is approximately half of the final IIEE error at day 10. This is rightly mentioned (L100) but this striking fact is not discussed further in Section 4 or the abstract. The authors note this is related to initialisation challenges and the use of only weakly coupled data assimilation. I know this is also a problem at other centres and is likely to be an issue for a number of years for coupled forecasts. I wonder if this finding should receive more prominence in the paper.

We agree and have made this more prominent by giving the comparison of the ocean5 analysis with MODIS fields its own subsection and mentioning the comparison of the initial and day-10 error in Section 4 and in the abstract. We have also extended the discussion of the issues with the DA as suggested. Following some internal discussions we also now mention that the minimisation of the sea ice concentration and other ocean state variables (i.e. T&S) is separate, which may be an important factor here. However, a more dedicated study of the DA procedure/performance around the polar and sub-polar seas will likely be necessary to identify the causes and come up with specific actions/improvements.

(3) The other related issue, which is briefly mentioned, is the veracity of the sea-ice analysis. The authors point out there are uncertainties in the sea-ice analyses and this will affect initialisation and the size of the errors (P10, L325) and that "guidance … from the remote sensing community" is needed. I agree here and I would perhaps suggest this limitation is added to the abstract. At present the last two lines of the abstract are a bit vague. It might be worth expanding these to state explicitly that the quality of satellite sea-ice products on daily to weekly timescales and on meso-scales (<500 km say) are not well characterised and this is a limitation for NWP.

We agree that this should be given more prominence. Have added the following to the abstract: "The importance of the choice of sea ice analysis for verification is also highlighted, with a call for more guidance on the suitability of satellite sea-ice products to verify forecasts on daily to weekly timescales and on meso-scales (<500 km)."

**Specific Comments**

L42 – there is another recent idealised modelling study on the atmospheric response to sea-ice geometry and concentration that should be cited here:

Spensberger, C., & Spengler, T. (2021). Sensitivity of air-sea heat exchange in

cold-air outbreaks to model resolution and sea-ice distribution. Journal of

Geophysical Research: Atmospheres, 126, e2020JD033610. https://doi.

org/10.1029/2020JD033610

Great! Have added this to the list.

L49 – I am not an expert on the timelines here, but are you sure that ECMWF developed the first coupled global … system? Maybe for an ensemble? Not sure about NWP more generally. The Canadian ECCC have had a coupled forecast model for some time and this may pre-date the ECMWF development. You cite one paper for the Canadian system (Smith et al. 2018), but you should probably also cite earlier pioneering work that demonstrated the potential for improvement in atmospheric forecasts from such a coupled system withi NWP.

Pellerin P, Ritchie H, Saucier SJ, Roy F, Desjardins S, Valin M, Lee V. 2004.Impact of a two-way coupling between an atmospheric and an ocean – icemodel over the Gulf of St. Lawrence.Mon. Weather Rev.132: 1379 – 1398

Smith GC, Roy F, Brasnett B. 2013. Evaluation of an operational ice-ocean analysis and forecasting system for the Gulf of St Lawrence. Q. J. R. Meteorol. Soc. 139: 419–433. DOI:10.1002/qj.1982

Smith, G.C., Roy, F., Reszka, M., Surcel Colan, D., He, Z., Deacu, D., Belanger, J.M., Skachko, S., Liu, Y., Dupont, F. and Lemieux, J.F., 2016. Sea ice forecast verification in the Canadian global ice ocean prediction system. Quarterly Journal of the Royal Meteorological Society, 142(695), pp.659-671.

We are quite sure this statement is correct. Although ECCC have been running coupled regional configurations for some time, a global coupled model was not implemented until Nov 2017: https://collaboration.cmc.ec.gc.ca/cmc/cmoi/product_guide/docs/tech_notes/technote_gdps-600_20171101_e.pdf. Whereas the ECMWF ensemble included coupling to both sea-ice and ocean since 2016. We have added a citation to the 2017 newsletter article describing this upgrade.

We have also added the following sentence to the paragraph to reflect these important coupled NWP studies mentioned. *"Indeed, pioneering efforts with coupled regional NWP systems have shown promising improvements in both*

*sea-ice and atmosphere fields compared to atmosphere-only systems (Pellerin et al., 2004 and Smith et al., 2013)."*

L79-80 – I was slightly confused on reading the explanation for the persisted surface conditions for the first time, because 'an anomaly is added each day'. On second reading I think this anomaly is only for the SST (not the sea ice)? Perhaps check for clarity here.

Modified for clarity as follows: "...one atmosphere-only where sea ice concentration and SST-anomalies are persisted from the initial time"

L128 – I think the Hersbach ERA5 reference is missing.

Added.

L142 – I'd replace "Atlantic coast' with Labrador Sea coast, as it isn't the main Atlantic basin.

Changed.

L231 – 'that region' – it is unclear from this paragraph which region you are talking about. Maybe these lines should be merged into the previous paragraph?

The paragraph was edited to include the region (Nordic Seas) explicitly in the text and added more references to Figure 7. We have also edited to actually mention the ice retreat composites, which are shown in the right-hand column of Figure 7, but were not actually mentioned in the text in the original submission.

L250-265 – this paragraph on internal boundary layer development at the ice edge is unreferenced – you could cite the Spensberger and Spengler 2021 paper here or the idealised 2D model of this internal BL development which also uses observations in

Renfrew, I.A. and King, J.C., 2000. A simple model of the convective internal boundary layer and its application to surface heat flux estimates within polynyas. Boundary-layer meteorology, 94(3), pp.335-356.

We have added a citation to both papers for the off-ice convective BL development situation and added a reference to Pithan et al. for the on-ice flow situation.

L285 – It was useful context to point out the differences in specific humidity (in g/kg and that this was 10% of the total value). You could also have expressed this as % of the standard deviation of this variable or something? And done similar for the

difference in temperature. I think it useful to have an idea of the magnitude of these forecast differences in the context of day to day variability. If you can easily do such a metric? This is just a suggestion, not necessary.

We agree that it would be nice to include this, but having tried it and found it very noisy, think it will require a bit more work to come up with something robust. So will opt to omit this from the revision.

L307 – The final section is more of a "Conclusions and Discussion" section.

Changed section heading as suggested

L355 "weakly" not weekly.

Changed as suggested.

**Figures**

Figure 1 – I would recommend changing the colour scale to one with white in the middle. At present the whole North Atlantic (which has no sea ice) is pink. It looks odd!

We have adjusted the scale so that it is white at the centre.

Fig 2 – the font size of the labels and legend is too small to read. Nice figure though!

The font sizes have been increased

Fig 3 & 2 – would it make sense to try and have the same colour for Ocean5 in these figures – this is red in 2 and 4 but green in fig 3.

Whilst we agree that it would be nice to use red for ocean5 here. Having tried this, we think the colour clashes with the red shading for snow/ice in corrected reflectance MODIS images colour (see below), so propose to stick with the original colour selection since we believe the figure is clearer in the original.

[Figure]

[Figure]

Fig R1: Fig 3a & b but with different line colours for the analysis ice edge.

Fig 5 – these figures illustrate the large variability between forecasts. Fig 5 is only very briefly mentioned in section 3.2 – I wonder if you should add a sentence or two emphasising the large variability.

*We have added the following to section 3.3: "However, the magnitude of the IIEE is highly variable from day to day, with the values of the IIEE varying by as much as an order of magnitude between the most and least skilful forecasts, depending on the region."*

Reviewer 2:

I enjoyed reading this article very much. The results are relevant for the scientific community, numerical weather prediction centers, and forecast users. The forecast verification process is supported by a solid and sophisticated methodological base, and the forecast improvements and deficiencies are honestly highlighted without over or understating the findings. Furthermore, the manuscript is well written, and the figures illustrate the outcome of the study appropriately. I include below a few remarks and suggestions, which are mostly minor, and I hope that these will help the authors in the revision process. I recommend the publication of with manuscript once those (minor) points are addressed.

GENERAL COMMENTS

The Introduction and Method sections provide a very good overview of the system. However, I think some details on the probabilistic nature of the forecast are missing. Underlining the higher compatibility of a coupled model configuration with the ECMWF ensemble forecast system (i.e. the ice can evolve independently in each

ensemble member, unlike in the persistence based strategy) would represent a nice addition to the study.

We have added the following at line 60: "*In the ensemble context this has the additional benefit that each ensemble member can have its own sea ice and SST fields consistent with the evolution of the meteorology. However, only deterministic forecasts will be considered in the present study.*"

What about the melting season? I expect the impact of the sea ice on the ocean and land weather to be limited in summer because of the milder temperature gradients and winds. However, the demand for good ice forecasts might peak during this season. I would briefly mention whether the features of the dynamical system are appropriate also for the summer months. I also think a brief reference to what happens to the Southern Ocean sea ice might be appropriate.

See the response to the following point.

The study focuses on a single winter season. Given the large number of forecasts analyzed, I expect the results to be solid. However, I think it might be good adding a characterization of the sea ice state during that winter in comparison to the climatological state, and discussing whether the results might be sensitive/influenced by potentially anomalous conditions (e.g. fast ice drift, abrupt melting events, etc.)

To address this and the previous point we have added the following paragraph to the final section: *"Geographically we have focussed our attention on the northern hemisphere in winter, since this is the time of year when the evolution of the sea ice has the largest influence on turbulent exchange and is therefore likely to have most relevance for forecasts of the atmosphere. However, the limitations of a persistence forecast of the ice edge and the benefits of dynamical forecasts for capturing the evolution are likely common to summer months and the southern hemisphere. Similarly, the analysis is limited to one season: DJFM 2017-18 which from a climatological perspective was quite unusual as many of the days were record lows for the time of year. However, this is unlikely to affect the general conclusions of the paper."*

This provides some context for the season in question with respect to the NH extent climatology and mentioned what we might expect to see in summer months. However, it is not clear how to provide a more event-based context in terms of abrupt melting events etc. without doing a separate detailed analysis, which we think is outside the scope of the paper.

Figure 4 clearly shows that OCEAN5 reanalysis is biased, and you describe this well in the text. However, I think giving some more context on the origin of this bias would be helpful for the readers.

We have also extended the discussion of the issues with the DA to provide some more context on the origin of the bias. In section 3.2 we now mention the separate

minimisation of the sea ice concentration and other ocean state variables (i.e. T&S) which may be an important factor here. However, a more dedicated study of the DA procedure will likely be necessary to identify the causes and come up with specific actions/improvements.

The fact that the thickness is not coupled implies that the thermodynamical transition at the ice edge is probably not well simulated also by the current dynamical system. Could you quantify the impact of this on the evolution of the internal boundary layer? Is the effect of a progressively reduced thickness towards the marginal ice zone negligible compared to the reduction in concentration? I expect this would also changes with the progressing of the freezing season. I think some more details on this in the discussion/conclusion section would be interesting for the reader.

This is a very interesting question and an important one for the design of future forecasting systems, however, the question is very difficult to answer using this set of experiments presented here. I would expect that such questions could be better answered using a more idealised experimental framework with prescribed sea ice conditions, such as in Spensberger, C., & Spengler, T. (2021).

I would like to point out that the using the AMSR2 derived sea ice concentration has also some drawbacks. It certainly comes with a desirable higher resolution because it uses higher frequencies. However, the effect of clouds on the microwave signal at higher frequencies is substantial and can penalize the quality of the retrieval, particularly across the marginal ice zone where clouds are not uncommon.

We have added a sentence to reflect this: *"However, is more affected by the presence of clouds than the Special Sensor Microwave Imager / Sounder (SSMIS)."*

SPECIFIC COMMENTS

Line 71: I would not consider obs-SSTSIC a real forecast but rather an hindcast or an AMIP type simulation.

We agree and actually none of these are forecasts, since they are simulations with a forecasting system for a period in the past. In this initial paragraph where the experiments are introduced, we call these "forecast experiments" and explain that they are produced with the IFS.

Line 75: I think it is worth mentioning here that the sea ice description of OSTIA comes from OSI-SAF. You mention this later in the result section, but I think stating this here would help the reader to understand the verification method.

The sentence from the following paragraph has been moved here.

Line 94: I suggest expressing the typical Arctic resolution of the ORCA025 grid also in km.

The typical resolution in km is now stated.

Line 96: Do you mean "...that is coupled to the atmosphere is..."?

Yes, thanks for pointing out the error.

FIGURES

Fig. 1: I suggest using a different colormap with the white color centered on zero. I don't like seeing the rectangular domain of the polar stereographic grid in pink. Using red and green for the boxes might not be color friendly.

We have adjusted the scale so that it is white at the centre and the green box was changed to orange.

Fig. 2: Labels and titles are too small. I think it is ok to lose the outlier point in the OSI-SAF timeseries (plots a and c), probably caused by a partial observational coverage on that day.

The font sizes have been increased and the outlier has been removed.

Fig. 4 and 6: Labels are too small, and I suggest using the scientific notation to improve the readability of the plots.

The font sizes have been increased and the units changed to millions of square km.